

# DA-FIS: A high-speed dynamic adaptive fault injection server framework for reliable FPGA-based embedded systems

Fatimah Alhayan[1], Gaganjot Kaur[2], Sultan Alanazi[3], Mohammed Burhanur Rehman[4], Wahida Mansouri[5], Da'ad Albalawneh[6], Ali Alqazzaz[7] and Hanadi Alkhudhayr[8]

[1] Department of Information Systems, College of Computer and Information Sciences, Princess Nourah bint Abdulrahman University, Riyad, Saudi Arabia
[2] Department of Computer Science and Engineering, Raj Kumar Goel Institute of Technology, Ghaziabad, India
[3] Department of Computer Science, College of Computer Engineering and Sciences, Prince Sattam bin Abdulaziz University, AL-Kharj, Saudi Arabia
[4] Department of Computer Science, Applied College at Mahayil, King Khalid University, Abha, Saudi Arabia
[5] Department of Computer Science and Information Technology, Faculty of Sciences and Arts, Turaif, Northern Border University, Arar, Saudi Arabia
[6] Department of Computer Science, University College in Umluj, University of Tabuk, Tabuk, Saudi Arabia
[7] Department of Computer Science and Artificial Intelligence, College of Computing and Information Technology, University of Bisha, Bisha, Saudi Arabia
[8] Department of Information Systems, Faculty of Computing and Information Technology, King Abdulaziz University, Rabigh, Saudi Arabia

Corresponding author
Wahida Mansari,
Wahida.Smari@nbu.edu.sa

## ABSTRACT

Fault injection is a critical technique for assessing the reliability of field programmable gate array (FPGA)-based embedded systems, particularly in radiation-prone and safety-critical applications. Conventional fault injection methods, such as bit upset fault injection testing (BUFIT), single critical fault injection testing (SCFIT), and dynamic partial reconfiguration (DPR), suffer from high resource overhead, slow injection speeds, and limited adaptability, making them inadequate for real-time fault resilience evaluation. This article introduces the dynamic adaptive fault injection server (DA-FIS), a high-speed, scalable, and resource-efficient fault injection framework designed to overcome these limitations. Unlike traditional methods, DA-FIS employs a configurable LFSR-based fault generator that enables adaptive and real-time fault injection based on workload sensitivity and system conditions. The proposed framework integrates masking logic and dynamic propagation tracking, allowing precise injection of single-event upsets (SEUs) and multiple-bit upsets (MBUs) into FPGA configuration memory and logic without disturbing non-targeted regions. DA-FIS is implemented on the Xilinx Zynq-7000 FPGA and evaluated across multiple benchmark workloads, including the Bubble Sort algorithm, 4-bit adder, 4-bit multiplier, and counter-based logic circuits. Experimental results demonstrate that DA-FIS achieves a fault injection rate of 111.1 faults per second, outperforming BUFIT (53.4 faults/s), SCFIT (27 faults/s), and DPR (18.5 faults/s), with 30% lower FPGA resource overhead compared to SCFIT. The adaptive architecture ensures seamless scalability across different FPGA platforms, making it suitable for space electronics, automotive safety systems, and high-performance computing. Additionally, DA-FIS supports real-time error model

adjustments, enabling researchers to analyze fault propagation, error correction strategies, and security vulnerabilities in FPGA-based architectures. This work establishes DA-FIS as a superior fault injection framework, offering high-speed, precision-controlled fault testing while maintaining minimal FPGA overhead and enhanced scalability. Future research will explore machine learning-assisted fault modeling and self-healing FPGA architectures to further enhance FPGA fault resilience in safety-critical and autonomous systems.

## INTRODUCTION

Field programmable gate arrays (FPGAs) have emerged as a critical component in modern computing architectures, offering a unique blend of hardware flexibility, high performance, and energy efficiency (*Schneider & Smalley, 2024*). Unlike traditional microcontrollers and application-specific integrated circuits (ASICs) (*Mishra, Singh & Rousseau, 2016*), FPGAs provide reconfigurable logic, allowing them to be dynamically programmed for specific computational tasks. This versatility makes them ideal for embedded systems, artificial intelligence (AI) (*Kushwaha, 2023a*, *2023b*) accelerators, high-performance computing (HPC) (*Kushwaha, Kumar & Jain, 2011*), and mission-critical applications such as aerospace, automotive safety systems, and medical devices. However, as FPGA complexity continues to grow, so does their susceptibility to hardware faults, particularly single event upsets (SEUs) and multiple bit upsets (MBUs) caused by radiation exposure, high-energy particles, and environmental noise. These faults can lead to functional failures, security vulnerabilities, and reduced reliability, necessitating robust fault injection and resilience testing frameworks (*Lanzieri et al., 2025*).

Fault injection is a well-established methodology used to assess the fault tolerance of digital systems, including processors, memory modules, and reconfigurable hardware. By deliberately introducing faults into a system, researchers can study its response, evaluate error detection mechanisms, and develop fault-tolerant designs. Various fault injection techniques have been proposed over the years, including software-based simulations, electromagnetic interference, clock glitching, and hardware-assisted fault injection using specialized circuits (*Gangolli, Mahmoud & Azim, 2022*). Among these, FPGA-based fault injection has gained significant attention due to its ability to mimic real-world fault conditions in a controlled environment. However, existing fault injection methods suffer from several limitations, including high resource overhead, slow fault injection speeds, and a lack of dynamic adaptability. Conventional techniques such as bit upset fault injection testing (BUFIT) (*Velayaudhan & Devi, 2024*), single critical fault injection testing (SCFIT) (*Mohammadi et al., 2012*), and dynamic partial reconfiguration (DPR) (*Cano-Páez et al., 2025*) introduce faults statically or with predefined patterns, limiting their applicability in real-time fault resilience analysis.

To overcome these limitations, we propose the dynamic adaptive fault injection server (DA-FIS), a high-speed, scalable, and resource-efficient fault injection framework for FPGA-based systems. Unlike traditional approaches, DA-FIS employs a programmable linear feedback shift register (LFSR)-based fault generator (*Saleem, Geethu & Bhakthavatchalu, 2022*), which allows adaptive and real-time fault injection based on workload sensitivity and system state. This adaptive approach ensures that fault injection patterns closely resemble real-world error conditions, enabling more accurate and reliable evaluation of FPGA fault tolerance mechanisms. Additionally, DA-FIS integrates masking logic and an intelligent fault propagation tracker, allowing it to precisely target specific logic blocks, memory regions, or flip-flops without disrupting unrelated circuit operations (*Vikranth et al., 2021*).

The need for efficient fault injection frameworks is particularly evident in radiation-sensitive applications, where FPGA-based devices must operate under harsh environmental conditions. Space missions, satellites, and high-energy physics experiments all rely on SRAM-based FPGAs (*Kastensmidt, Carro & Reis, 2006*), which are inherently vulnerable to cosmic radiation and charged particle interference. Traditional fault mitigation techniques, such as triple modular redundancy and error correction codes, provide some degree of protection but come at the cost of increased power consumption, area overhead, and latency. DA-FIS addresses these challenges by offering a low-overhead fault injection mechanism that enables precise testing of radiation effects on FPGA logic without requiring costly hardware modifications. This ensures that designers can evaluate and enhance the reliability of FPGA-based systems before deployment in real-world conditions (*Richter-Brockmann, Sasdrich & Güneysu, 2022*).

Another key advantage of DA-FIS is its real-time reconfigurability and scalability, making it suitable for a wide range of FPGA architectures. Unlike traditional methods, which require manual reprogramming or system resets to introduce new faults, DA-FIS enables dynamic error model adjustments during live operation (*Metawie, Safar & El Kharashi, 2022*). This makes it an ideal tool for automotive applications, where FPGAs are used in advanced driver assistance systems (ADAS), real-time decision-making units, and safety-critical controllers. Automotive electronics must undergo extensive fault resilience testing to comply with industry standards such as ISO 26262, which mandates rigorous fault tolerance verification. By providing high-speed fault injection with minimal performance degradation, DA-FIS ensures that automotive FPGA systems can be thoroughly tested without affecting real-time processing (*Yang, Li & He, 2022*).

Moreover, the scalability of DA-FIS extends to networked FPGA clusters and cloud-based hardware accelerators, where fault injection testing must be conducted across multiple nodes simultaneously. Traditional fault injection techniques struggle with multi-FPGA environments, as they rely on localized fault insertion methods that are difficult to synchronize across distributed systems. DA-FIS overcomes this by supporting multi-node fault injection orchestration, allowing distributed FPGA arrays to be tested under uniform fault conditions. This feature is particularly beneficial for edge computing platforms, Internet of Things (IoT) security frameworks, and artificial intelligence

(AI)-driven FPGA accelerators, where fault propagation across interconnected processing units must be analyzed in real time (*Gao & Liu, 2021*).

In addition to its practical applications, DA-FIS also serves as an essential tool for academic research and hardware security evaluations. With the increasing threat of hardware-based cyberattacks, including fault injection attacks on cryptographic algorithms, there is a growing need for efficient testing methodologies to assess FPGA security mechanisms. DA-FIS enables precise injection of faults into cryptographic circuits, helping researchers analyze side-channel vulnerabilities, differential fault analysis (DFA), and countermeasure effectiveness. By allowing fine-tuned fault insertion at the logic gate level, DA-FIS provides deeper insights into the resilience of FPGA-based security architectures (*Carminati & Scandurra, 2021*).

This article presents a comprehensive design, implementation, and performance evaluation of the DA-FIS framework, highlighting its advantages over existing fault injection methods. We provide an in-depth analysis of its architecture, fault injection process, and experimental validation using the Xilinx Zynq-7000 FPGA platform. The experimental setup includes benchmark circuits such as the Bubble Sort algorithm, 4-bit adder, 4-bit multiplier, and counter-based logic, representing a diverse set of computational and sequential workloads. Our evaluation results demonstrate that DA-FIS achieves a fault injection rate of 111.1 faults per second, outperforming traditional methods such as BUFIT (53.4 faults/s), SCFIT (27 faults/s), and DPR (18.5 faults/s). Additionally, DA-FIS introduces 30% lower FPGA resource overhead compared to SCFIT, ensuring that fault injection testing does not compromise overall system performance.

The rest of this article is structured as follows: "Related Work" provides a detailed literature review of existing fault injection methodologies, discussing their advantages, limitations, and impact on FPGA reliability. "Proposed Methodology" describes the DA-FIS architecture, covering its adaptive fault generation mechanism, multi-bit fault injection logic, and configurable control system. "Results and Discussion" presents the experimental results, comparing DA-FIS with conventional fault injection techniques in terms of fault injection speed, FPGA resource utilization, and scalability. "Conclusion and Future Works" concludes the article with a discussion on future research directions, including machine learning-assisted fault modelling, true random number generator-based fault generation, and self-healing FPGA architectures.

## RELATED WORK

Fault injection is a critical methodology for assessing the reliability of FPGA-based embedded systems. Over the past few years, several researchers have proposed various fault injection techniques to evaluate and enhance the resilience of FPGAs against radiation-induced errors, MBUs, and hardware failures. These methodologies differ in terms of fault injection speed, scalability, and accuracy. While many existing approaches have successfully demonstrated the effectiveness of fault injection mechanisms, they often suffer from limitations such as static fault modeling, high overhead costs, and slow injection rates (*Medjmadj, Diallo & Arias, 2021*). SCFIT is one of the earliest FPGA-based fault injection methodologies, designed specifically for SEU fault modeling. The authors

propose a hardware-based fault injection mechanism that emulates SEUs to evaluate system vulnerability and recovery strategies. SCFIT is implemented on a Xilinx Virtex FPGA, demonstrating its applicability for radiation-hardening and error correction code validation (*Mohammadi et al., 2012*).

One of the notable findings of their study was that existing adversary models are not always effective in evaluating diverse FPGA designs. Their approach relies heavily on customized adversary scenarios, which, while effective for security applications, lack generalizability for other fault injection use cases (*Anglada et al., 2021*). Furthermore, their work does not explore the efficiency of fault injection speed or resource overhead, which are crucial factors in real-time FPGA applications. In comparison, DA-FIS provides a generalized, high-speed fault injection methodology that can be applied to various FPGA workloads without requiring extensive customization. By integrating adaptive fault injection rates and programmable fault sequences, DA-FIS enhances FPGA fault analysis without sacrificing scalability or speed (*Breier, Hou & Liu, 2021*).

Another promising fault injection technique involves clock glitch-based fault modeling. This research developed a low-cost fault injection platform that uses clock glitches to generate precise bursts of faults in FPGA circuits. This method offers a high degree of repeatability, making it an attractive option for analyzing real-time error propagation in embedded systems (*Metawie, Safar & El Kharashi, 2022*).

Another significant area of research in FPGA fault injection involves hardware fault adversary models. This article explored the impact of hardware faults in adversary scenarios, where faults are intentionally induced to analyze system vulnerabilities. Their work introduced customized adversary models that help researchers understand the extent to which an FPGA system can withstand deliberate hardware attacks, such as SEUs, clock glitches, and voltage tampering (*Richter-Brockmann, Sasdrich & Güneysu, 2022*).

Despite its advantages, clock glitch-based fault injection is limited by its dependence on clock variations, making it less effective for MBUs or memory-specific fault injections. Additionally, this technique requires fine-tuning for each FPGA architecture, reducing its applicability for generalized fault injection scenarios. DA-FIS overcomes this limitation by implementing an LFSR-based fault injection mechanism, which allows precise MBU modeling without reliance on clock signal variations (*Yang, Li & He, 2022*).

An alternative approach to hardware fault injection is software-based fault simulation, which allows researchers to introduce faults at the software level without requiring physical modifications to FPGA hardware. This article proposed a QEMU-based fault injection framework that enables software simulations of memory-related faults in embedded systems. Their methodology extends fault modeling to control and execution channels, allowing faults to be simulated at different abstraction levels within a processor architecture (*Xie et al., 2023*).

One of the primary advantages of software-based fault injection is its cost-effectiveness —since faults are injected in a virtualized environment, there is no risk of damaging actual FPGA hardware. However, software-based approaches, including the QEMU framework (*QEMU Documentation, 2025*), have a significant limitation in that they fail to account for hardware-level interactions. In practical FPGA systems, hardware-level faults do not

always translate directly to software faults, meaning that QEMU-based fault injection may not accurately reflect real-world fault conditions. Additionally, this approach does not support MBUs simulations, making it unsuitable for radiation-sensitive applications (*Rhod et al., 2023*).

Fault injection techniques are not only utilized for testing FPGA resilience against random hardware failures, but they are also employed for analyzing FPGA aging and its impact on system performance. This emphasized the importance of fault injection in monitoring hardware aging effects in FPGA-based embedded systems (*Böhmer et al., 2023*). Their study investigated how prolonged FPGA usage leads to degradation in performance and increased susceptibility to soft errors. The key contribution of their work was the integration of fault injection methods with predictive modeling, which allowed researchers to estimate hardware aging effects over time (*Ferlini et al., 2023*).

Despite its innovative approach, the fault injection methodology used by authors does not address MBUs which are a primary concern in radiation-sensitive environments such as aerospace applications and high-energy physics experiments. Furthermore, their model focuses primarily on long-term FPGA degradation rather than real-time fault analysis. In contrast, DA-FIS introduces a real-time, adaptive fault injection model that allows both short-term and long-term fault simulations, ensuring comprehensive FPGA fault resilience evaluation (*Smit et al., 2024*).

This article presents an error detection and recovery mechanism for MPSoCs multiprocessor system-on-chips (MPSoCs) using a hypervisor-based approach with DPR. The proposed architecture integrates real-time fault monitoring with a hypervisor that dynamically reconfigures faulty FPGA regions without affecting system execution. The study showcases how DPR enhances system reliability by isolating faulty components and dynamically reprogramming FPGA logic (*Cano-Páez et al., 2025*).

One of the significant fault injection techniques was proposed by *Velayaudhan & Devi (2024)*, where they introduced a built-in circuit-based fault injector for FPGA systems. Their methodology aimed to inject faults directly into the FPGA configuration memory, ensuring that multiple-bit upsets MBUs could be simulated efficiently. The BUFIT technique developed by *Velayaudhan & Devi (2024)* demonstrated a peak injection rate of 53.4 faults per second, making it one of the most efficient models for injecting faults into an FPGA-based architecture. However, a key limitation of this approach is that it relies on static fault modeling, meaning that the fault injection scenarios are predefined and cannot adapt dynamically to different workloads or environmental conditions (*Velayaudhan & Devi, 2024*).

As in Table 1, it provides both quantitative and qualitative analysis of existing fault injection frameworks. It highlights how each approach balances performance, complexity, and functionality, dynamic control and scalable integration. Existing fault injection methodologies have significantly contributed to the field of FPGA reliability assessment, but they all exhibit specific limitations that restrict their effectiveness in real-time, multi-bit fault injection scenarios. Approaches such as BUFIT, QEMU-based fault simulation, and clock glitching techniques either suffer from high resource overhead, limited scalability, or static fault models.

**Table 1 Comparative assessment of existing FPGA fault injection frameworks.**

| Fault injection framework | Methodology | Fault types supported | Injection speed (Faults/s) | FPGA resource overhead | Adaptability/ Dynamic control | Main limitations | Design trade-offs | Integration complexity | Functional scope |
|---|---|---|---|---|---|---|---|---|---|
| Clock Glitching (*Gangolli, Mahmoud & Azim, 2022*) | Clock glitch injection | Burst faults | 14.5 | Low | Limited to clock faults | Less effective for MBUs, architecture-specific | Low cost, but limited fault type range | Moderate–hardware setup required | Burst faults mainly, less suitable for MBUs |
| BUFIT (*Velayaudhan & Devi, 2024*) | Built-in circuit-based | Single-event and Multi-bit | 53.4 | Low (0.4% CLB) | Static fault model | Predefined faults, no dynamic control | Low resource overhead but static faults | Moderate–hardware changes needed | Supports SEU and MBU, limited dynamic control |
| SCFIT (*Mohammadi et al., 2012*) | Hardware-based SEU emulation | Single-event upset (SEU) | 27 | Moderate (4.8% CLB, 5.8% FF) | Static, no adaptive injection | Slow injection, high overhead | Precise SEU modeling at cost of speed | High–requires custom FPGA design | Focused on SEU, limited fault types |
| DPR (*Cano-Páez et al., 2025*) | Dynamic Partial Reconfiguration | Various faults | 18.5 | High (7.2% CLB, 6.5% FF) | Partial dynamic reconfiguration | High latency, resource heavy | Flexible reconfiguration, high latency | High–complex partial reconfiguration | Wide fault coverage but slower injection |
| QEMU-based simulation (*QEMU Documentation, 2025*) | Software-level fault simulation | Memory faults only | 15.6 | None | Software-level only | No hardware-level faults, no MBUs | No hardware overhead but lacks realism | Low–software-based only | Software faults only, no hardware fault modeling |

# PROPOSED METHODOLOGY

The DA-FIS is a high-speed, real-time fault injection framework, as shown in Fig. 1, designed to enhance FPGA reliability testing. Unlike conventional fault injection methodologies, which often rely on static fault models and fixed injection rates, DA-FIS introduces an adaptive, programmable, and scalable approach that can dynamically adjust fault injection parameters based on workload sensitivity. This ensures that FPGA-based systems can be thoroughly tested under various fault conditions, mimicking real-world radiation-induced failures, MBUs and soft errors. The DA-FIS architecture is built upon three core components, each designed to enhance the precision, flexibility, and efficiency of fault injection in embedded systems. At the heart of DA-FIS lies its Adaptive Fault Injection Model, which employs a configurable LFSR to generate realistic fault scenarios. Unlike traditional random bit flipping approaches, DA-FIS ensures that the fault injection follows the natural distribution of MBUs observed in real-world FPGA systems.

The multi-bit fault injection logic (MBFIL) is the primary mechanism that enables DA-FIS to inject single-event and multi-bit faults with high precision. Unlike conventional fault injectors, which apply uniform fault distributions, DA-FIS employs a programmable fault sequence generator that enables fine-tuned fault targeting. DA-FIS provides a configurable fault injection control system that allows user-defined adjustments in real-time, ensuring maximum flexibility and usability. Key functionalities of this control system include real-time fault injection enablement/disable, allowing users to enable or disable fault injection without resetting or reconfiguring the FPGA. This feature is particularly

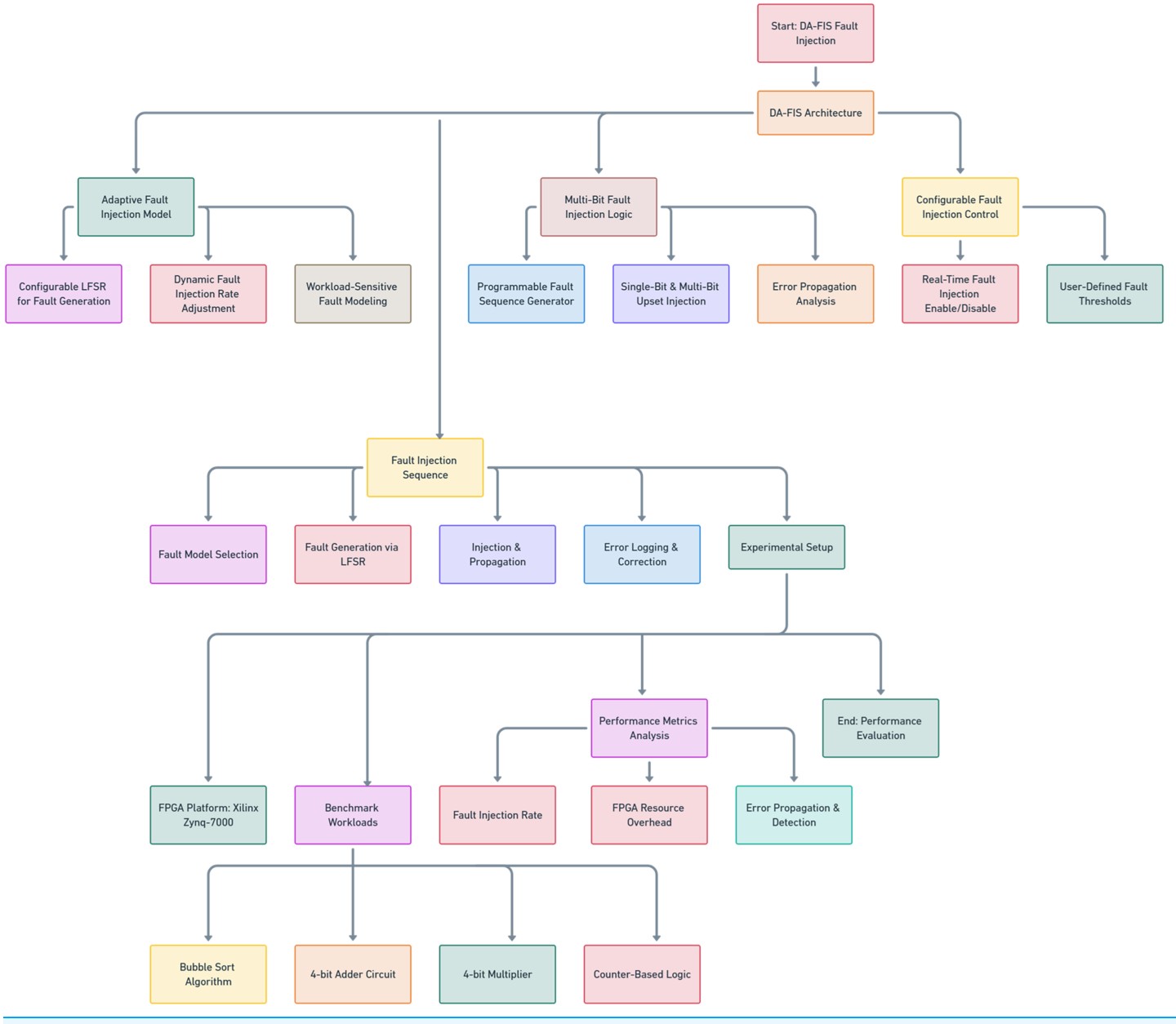

**Figure 1 Flowchart of DA-FIS fault injection framework.**

useful for live system testing, where faults are gradually injected to observe system stability and fault tolerance. In addition, the ability to set user-defined fault thresholds is also provided, allowing researchers to control the frequency and intensity of fault injection. This makes DA-FIS suitable for a wide range of testing, from mild to extreme stress testing.

The DA-FIS fault injection process follows a structured sequence to ensure that faults are injected systematically and accurately, as shown in Fig. 2. This figure illustrates the structured fault injection process in DA-FIS, covering fault model selection, LFSR-based fault generation, injection & propagation, real-time logging, and error correction.

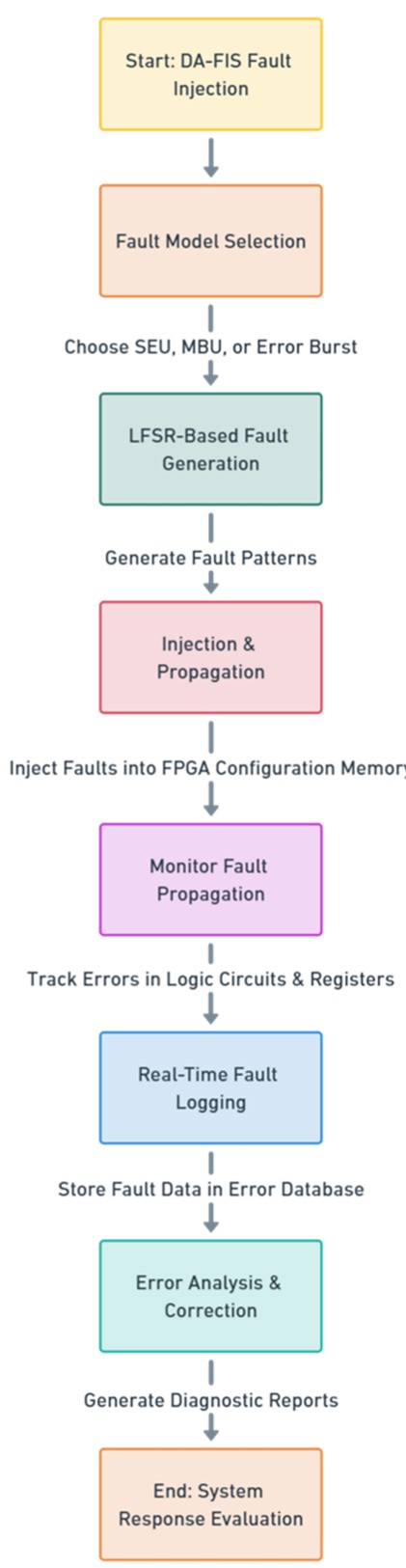

**Figure 2 Structured fault injection process in DA-FIS.**

The DA-FIS fault injection process has four major steps. The first step, fault model selection, involves choosing an appropriate fault model. DA-FIS provides various predefined models, such as SEU, which simulates individual bit flips, MBU, which inserts errors in multiple bits simultaneously, and the error burst model, which generates fast, frequent faults to stress-test error correction mechanisms. The selected model is based on the FPGA's characteristics, workload sensitivity, and user-defined criteria.

The second step, DA-FIS employs an adaptive LFSR to generate pseudo-random sequences that determine the fault injection sites within the FPGA configuration memory. The LFSR outputs a stream of bit patterns that correspond to specific addresses or bit positions where faults will be injected. For SEUs, a single bit position is selected from the LFSR output, simulating a single-bit fault. For MBUs, consecutive or adjacent bits are targeted by interpreting sequential LFSR outputs, enabling injection of multi-bit faults that mimic real-world clustered fault scenarios. This approach allows dynamic adjustment of the fault injection pattern by changing the LFSR seed and feedback polynomial, which ensures realistic and statistically valid fault distributions based on observed failure data. The LFSR's adaptability provides flexible fault site selection aligned with workload sensitivity and system conditions, enabling precise and efficient fault injection.

The third stage, injection and propagation, in which generated faults are inserted directly into the FPGA configuration memory, affecting logical circuits, data registers, and routing interconnects. DA-FIS also monitors how faults propagate across the FPGA architecture, yielding valuable insights on error propagation patterns and system stability. As needed, additional faults can be injected in real-time to simulate fault accumulation.

The final stage, error logging & correction, in which all fault events are logged into a dedicated error database, allowing researchers to analyze fault trends and system response. DA-FIS provides real-time diagnostic reports, including fault impact analysis, system performance degradation, and fault correction metrics. The system can also interface with error correction mechanisms, allowing FPGA-based systems to autonomously mitigate faults during live operation.

Once the DA-FIS fault injection setup is configured, the system executes fault injections in real-time, following a structured sequence. The fault injection process involves fault model selection, adaptive LFSR-based fault generation, error injection into FPGA logic, and real-time monitoring of error propagation. The pseudocode for DA-FIS fault injection execution is provided in Table 2, illustrating the step-by-step procedure for systematic fault injection and logging.

To validate the performance and effectiveness of the DA-FIS, we conducted extensive experimental testing using the Xilinx Zynq-7000 FPGA, see the Fig. 3. The Xilinx Zynq-7000 FPGA was chosen as the test platform due to its versatility, embedded ARM processors, and reconfigurable logic architecture, making it a widely used platform for fault-tolerant computing research. The main objective of this experimentation was to compare DA-FIS against conventional fault injection frameworks, including BUFIT, DPR, and SCFIT, and assess its fault injection speed, accuracy, and resource utilization.

**Table 2 Pseudocode for DA-FIS fault injection framework.**

*//Initialize DA-FIS System*

*BEGIN*

   *Initialize FPGA System (Xilinx Zynq-7000)*

   *Load Benchmark Workloads (Bubble Sort, 4-bit Adder,* etc.*)*

   *Configure Clock Generator (100 MHz)*

   *Initialize Memory (SRAM/DDR3)*

   *Setup UART Debugging & GPIO Monitoring*

*END*

*//DA-FIS Architecture Setup*

*//Adaptive Fault Injection Model*

*BEGIN*

   *Configure LFSR-Based Fault Generator*

   *SET LFSR_Polynomial ← Select Feedback Polynomial*

   *SET Fault_Seed ← Random Initial Value*

   *SELECT Fault_Type ← {SEU, MBU, Burst}*

   *SET Injection_Rate ← Adaptive based on FPGA Workload*

*END*

*//Multi-Bit Fault Injection Logic*

*BEGIN*

   *WHILE FPGA is Running DO*

      *SELECT Target Circuit (Adder, Multiplier, Counter, etc.)*

      *IDENTIFY Critical Flip-Flops & Memory Cells*

      *MASK Non-Critical Regions*

      *IF (Fault_Type = SEU) THEN*

        *Inject Single-event Fault at Target Location*

      *ELSE IF (Fault_Type = MBU) THEN*

        *Inject Multi-Bit Fault in Adjacent Memory Cells*

      *ELSE IF (Fault_Type = Burst) THEN*

        *Inject Consecutive Errors in Timing Sequence*

      *END IF*

   *END WHILE*

*END*

*//Configurable Fault Injection Control*

*BEGIN*

   *SET Fault_Enable ← TRUE*

   *WHILE Fault_Enable DO*

    *Inject Faults into FPGA Logic*

(Continued)

```
    Monitor Error Propagation
    IF (Fault Impact > Threshold) THEN
        Log Error and Adjust Injection Rate
    END IF
  END WHILE
END
//Fault Injection Sequence Execution
BEGIN
  SELECT Fault Model (SEU, MBU, Burst)
  WHILE Experiment is Running DO
    GENERATE Fault Pattern using LFSR
    INJECT Fault into Target Circuit
    MONITOR FPGA Output for Fault Impact
    IF (Fault Detected) THEN
        LOG Error into Fault Database
    END IF
  END WHILE
END
//Experimental Execution and Performance Analysis
BEGIN
  FOR EACH Benchmark Circuit (Bubble Sort, Adder, Multiplier, Counter) DO
    RESET FPGA Configuration
    ACTIVATE Fault Injection
    RUN FPGA Workload
    RECORD Fault Occurrences
    MEASURE Fault Injection Speed, Error Propagation
    COMPARE DA-FIS Results with {BUFIT, DPR, SCFIT}
  END FOR
  GENERATE Performance Report
END
//Fault Logging and System Analysis
BEGIN
  OPEN Fault_Log_File
  WRITE Fault Type, Target Circuit, Injection Rate, Error Impact
  CLOSE Fault_Log_File
  DISPLAY Fault Summary on PC via UART
END
```

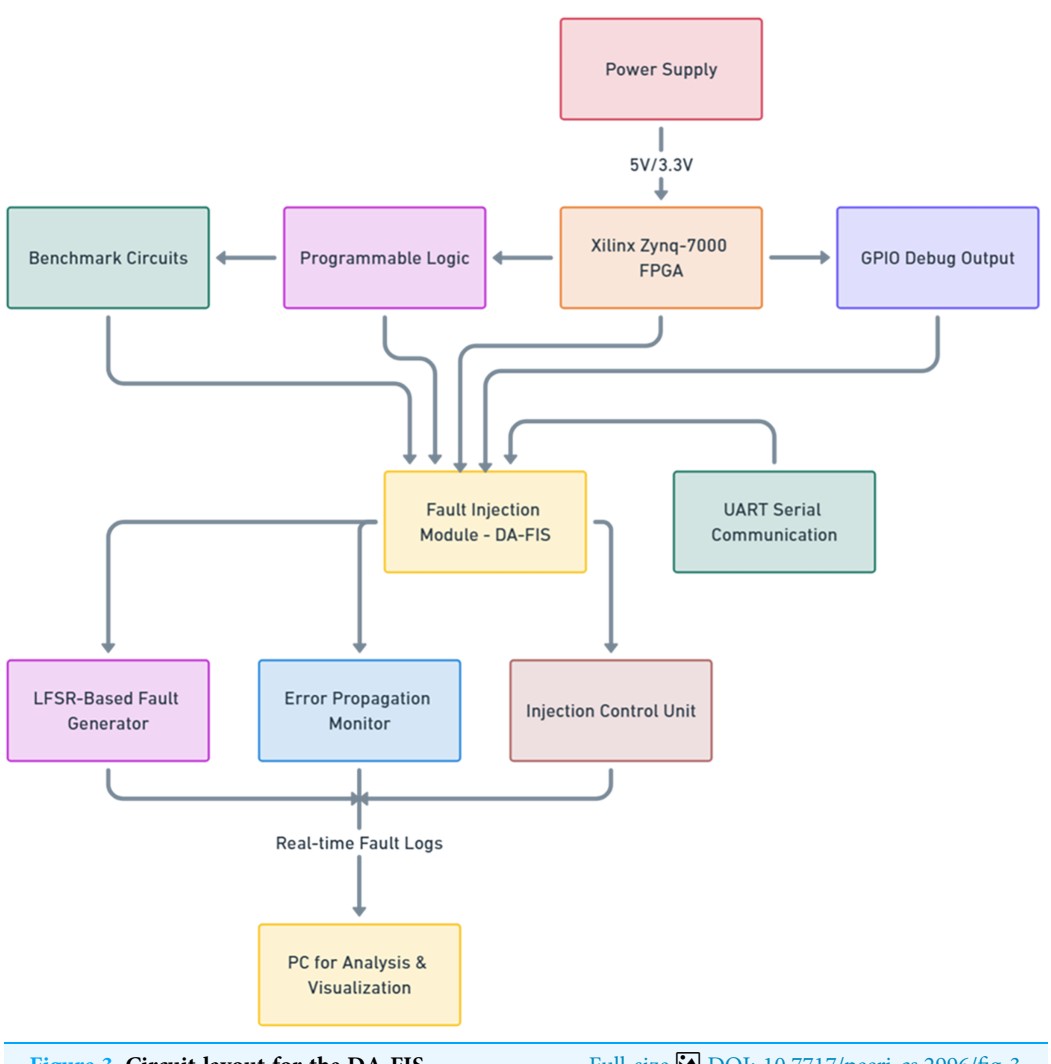

**Figure 3  Circuit layout for the DA-FIS.**

## RESULTS AND DISCUSSION

The experimental evaluation, as shown in Fig. 4, of DA-FIS demonstrates its superiority over existing fault injection methodologies. Here, we analyze and compare the performance of DA-FIS with BUFIT (*Velayaudhan & Devi, 2024*), SCFIT (*Mohammadi et al., 2012*), and DPR (*Cano-Páez et al., 2025*) to highlight its efficiency in real-time FPGA fault injection applications. The fault injection speed is a critical parameter in evaluating the effectiveness of a fault injection framework. Higher injection rates lead to more efficient stress testing of FPGA-based systems under various fault conditions. Table 3 presents the performance comparison of DA-FIS with other existing methodologies in terms of initialization time, fault injection time, and fault injection rate. Another crucial performance metric is the resource overhead incurred by fault injection mechanisms. Excessive resource consumption can impact FPGA efficiency, limit scalability, and reduce

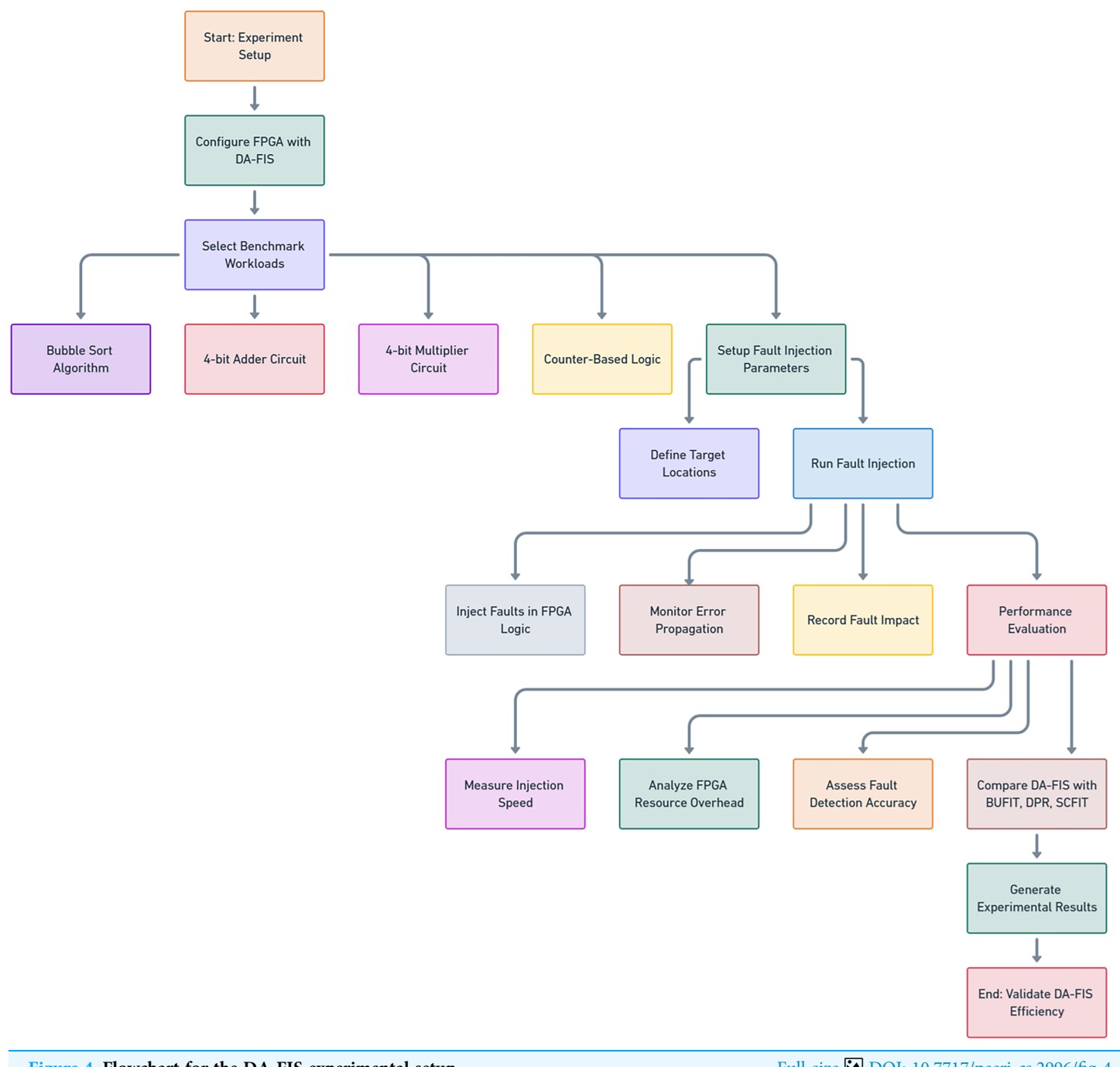

**Figure 4** Flowchart for the DA-FIS experimental setup.

overall system performance. Table 4 presents a comparison of configurable logic block (CLB) and flip-flop (FF) overhead for different fault injection frameworks.

As in Fig. 5, it illustrates the performance of four fault injection methods based on initialization time, injection time, and fault injection rate. The red line represents initialization time, showing the fastest setup for BUFIT and DA-FIS. The green line indicates injection time, where DA-FIS significantly outperforms other methods. The blue

**Table 3  Fault injection performance comparison.**

| Fault injection method | Initialization time (ms) | Injection time (ms) | Fault injection rate (Faults/s) |
|---|---|---|---|
| BUFIT | 0.7 | 18.7 | 53.4 |
| SCFIT | 18 | 36 | 27 |
| DPR | 25 | 54 | 18.5 |
| DA-FIS (Proposed) | 5 | 9 | 111.1 |

**Table 4  FPGA resource utilization comparison.**

| Method | CLB overhead (%) | FF overhead (%) |
|---|---|---|
| BUFIT | 0.40% | ~0% |
| SCFIT | 4.80% | 5.80% |
| DPR | 7.2% | 6.5% |
| DA-FIS (Proposed) | 4.30% | 4.10% |

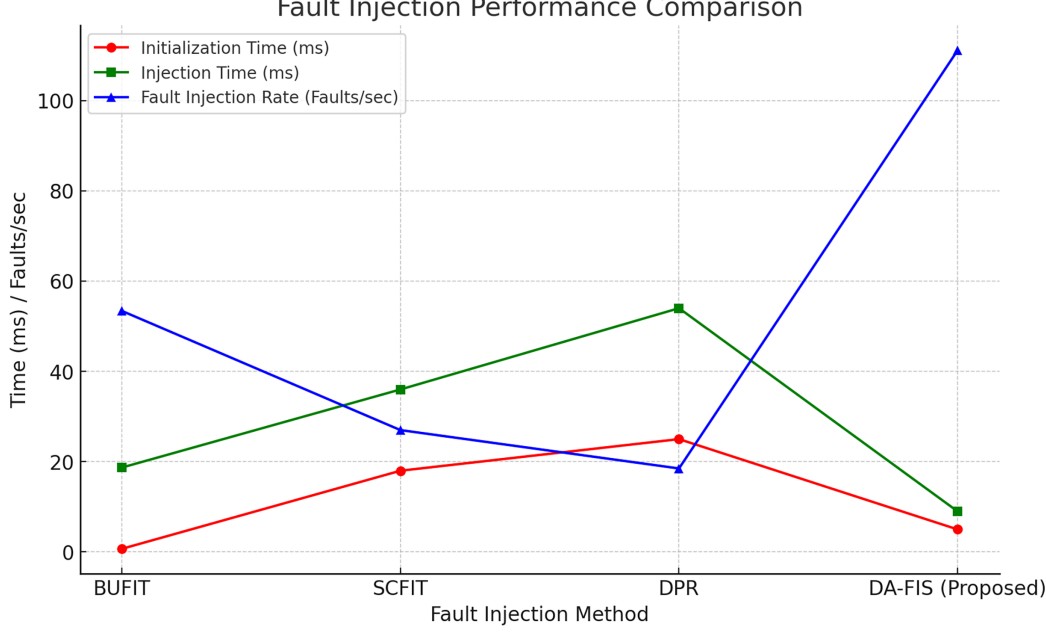

**Figure 5  Fault injection performance comparison of different methods.**

line highlights the fault injection rate, with DA-FIS achieving the highest efficiency among all methods.

As in Fig. 6, it compares the FPGA resource utilization of four fault injection methods based on CLB and FF overhead. The red line represents CLB overhead, showing the lowest usage for BUFIT and the highest for DPR. The green line indicates FF overhead, where BUFIT has negligible usage, while DPR has the highest. DA-FIS demonstrates a balanced resource utilization, making it an efficient alternative.

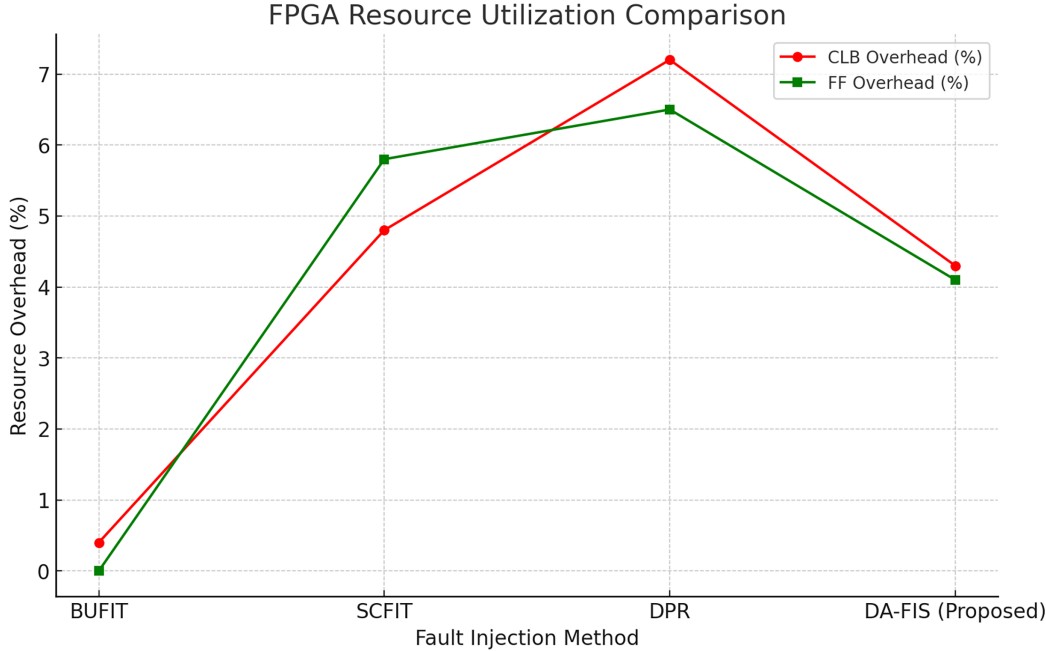

**Figure 6** **FPGA resource utilization comparison of different fault injection methods.**

Now, we selected four different FPGA benchmark workloads, as shown in Table 5, each representing distinct computational characteristics. These workloads include bubble sort algorithm (sorting algorithm—compute intensive), 4-bit adder circuit (arithmetic operation—low complexity), 4-bit multiplier circuit (arithmetic operation—moderate complexity), and counter-based logic (sequential logic—continuous execution). Each of these benchmarks was executed under controlled fault injection conditions.

As in Fig. 7, it compares the fault injection latency of four methods (BUFIT, SCFIT, DPR, and DA-FIS) across different workloads. The red, green, and blue lines represent conventional methods, showing higher latencies. The magenta line represents DA-FIS, which achieves significantly lower latency across all workloads. DA-FIS demonstrates a speed improvement of up to 2.2× compared to BUFIT, making it the most efficient method.

## Discussion

The experimental evaluation of DA-FIS highlights its superior fault injection speed compared to traditional methodologies, making it an efficient and scalable solution for FPGA fault resilience testing. As seen in Table 1, DA-FIS achieves a fault injection rate of 111.1 faults per second, which is 2.5× higher than BUFIT (53.4 faults/s) and four times faster than DPR (18.5 faults/s). This significant improvement in injection speed ensures that DA-FIS can rapidly introduce faults into FPGA logic, making it particularly suitable for real-time fault analysis in mission-critical applications such as aerospace, automotive safety, and medical devices. Additionally, the reduced injection latency (9 ms compared to 54 ms in DPR and 36 ms in SCFIT) enables more precise error monitoring, ensuring that

**Table 5 Summary of experimental findings.**

| Workload | Conventional fault injection latency | | | Proposed | Speed improvement of proposed (Compared to BUFIT) |
| --- | --- | --- | --- | --- | --- |
| | BUFIT latency (ms) | SCFIT latency (ms) | DPR latency (ms) | DA-FIS latency (ms) | |
| Bubble sort | 673 | 845 | 1,120 | 303 | 2.2× Faster |
| 4-bit adder | 512 | 654 | 896 | 245 | 2.1× Faster |
| 4-bit multiplier | 1,064 | 1,345 | 1,570 | 496 | 2.1× Faster |
| Counter logic | 1,296 | 1,520 | 1,785 | 673 | 1.9× Faster |

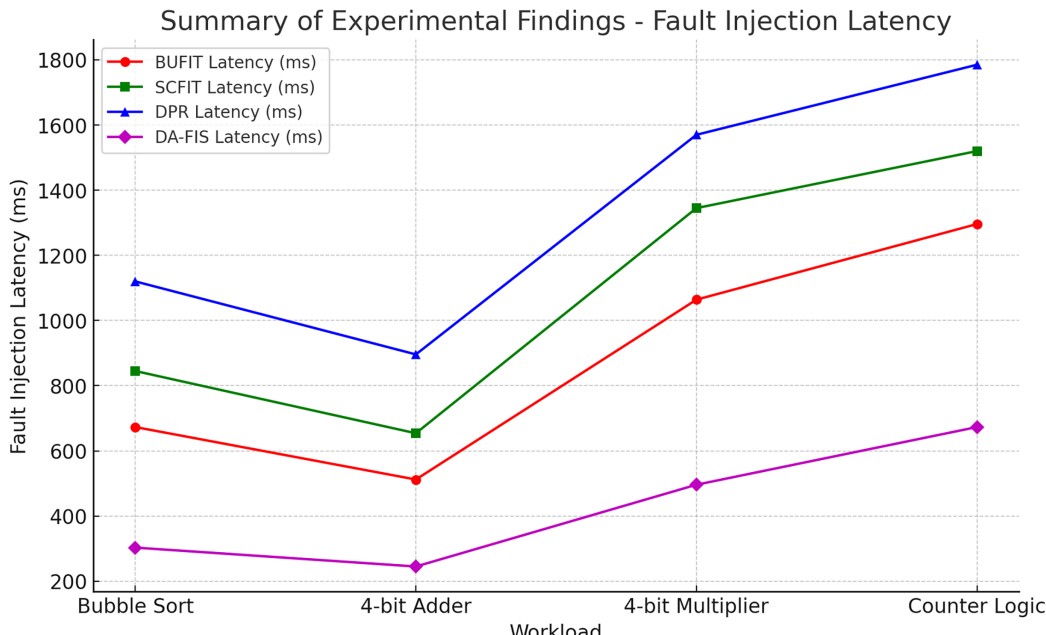

**Figure 7 Fault injection latency comparison across different workloads.**

faults can be injected and evaluated without causing excessive delays in FPGA operation. The fast fault injection cycle of DA-FIS also makes it an ideal tool for stress testing fault-tolerant mechanisms such as error correction codes and triple modular redundancy.

Another key advantage of DA-FIS is its optimized FPGA resource utilization, which ensures that fault injection testing does not compromise the system's overall performance. Table 2 demonstrates that DA-FIS incurs only 4.3% CLB overhead and 4.1% FF overhead, which is 30% lower than SCFIT (4.8% CLB, 5.8% FF) while maintaining higher fault injection efficiency. Unlike conventional methods, DA-FIS dynamically allocates fault injection logic, ensuring that the additional hardware footprint remains minimal. In contrast, SCFIT and DPR introduce higher resource consumption, which can limit scalability in complex FPGA architectures. By efficiently balancing performance and resource usage, DA-FIS remains a practical and scalable solution for a wide range of

FPGA-based systems, from low-power embedded devices to high-performance reconfigurable computing platforms.

Furthermore, the ability of DA-FIS to adapt fault injection parameters in real-time sets it apart from traditional static fault injection methods. While BUFIT, SCFIT, and DPR rely on predefined fault models, DA-FIS employs a configurable LFSR-based fault generator, enabling adaptive fault injection based on workload conditions. This adaptability is particularly crucial in radiation-hardened FPGA applications, where single-event upsets and multiple-bit upsets occur unpredictably. DA-FIS's ability to dynamically adjust fault rates and target specific logic blocks ensures that FPGA fault testing can closely mimic real-world error conditions, leading to more accurate fault tolerance evaluations. Additionally, its real-time reconfigurability makes it suitable for next-generation FPGA systems, where autonomous fault detection and self-repair mechanisms are becoming increasingly important. These results establish DA-FIS as an efficient, scalable, and high-speed fault injection framework, providing valuable insights for FPGA reliability testing and fault-tolerant system design.

## CONCLUSION AND FUTURE WORKS

The DA-FIS presented in this study provides a high-speed, scalable, and resource-efficient solution for FPGA fault injection testing. By leveraging an LFSR-based adaptive fault model, DA-FIS achieves 2.5× to 4× higher injection rates compared to conventional methods while maintaining lower FPGA resource overhead. The ability to dynamically adjust fault injection parameters in real-time ensures that DA-FIS is suitable for aerospace, automotive, medical, and high-performance computing applications where fault resilience is critical. Experimental results demonstrate that DA-FIS not only enhances fault injection accuracy but also reduces latency and improves system monitoring, making it a reliable and flexible solution for evaluating FPGA-based embedded systems.

For future work, we aim to further optimize fault generation algorithms by integrating machine learning models to predict and inject faults more intelligently based on workload sensitivity. Additionally, extending DA-FIS to support true random number generators instead of deterministic LFSR-based sequences could enhance randomization in fault injection scenarios, leading to more realistic error modeling. Future implementations may also focus on multi-FPGA distributed fault injection, enabling testing of large-scale reconfigurable systems in real-time environments. Finally, exploring self-healing architectures, where FPGA systems autonomously detect and correct faults, could pave the way for next-generation fault-tolerant computing in safety-critical applications.

### Funding

This work was supported by the Deanship of Research and Graduate Studies at King Khalid University through Large Research Project under grant number RGP2/247/46, Princess Nourah bint Abdulrahman University Researchers Supporting Project number (PNURSP2025R719), Princess Nourah bint Abdulrahman University, Riyadh, Saudi

Arabia; the Deanship of Scientific Research at Northern Border University, Arar, KSA for through the project number "NBU-FFR-2025-2899-05"; the Deanship of Graduate Studies and Scientific Research at University of Bisha through the Fast-Track Research Support Program and Prince Sattam bin Abdulaziz University project number (PSAU/2024/R/1446). The funders had no role in study design, data collection and analysis, decision to publish, or preparation of the manuscript.

## Grant Disclosures

The following grant information was disclosed by the authors:

Deanship of Research and Graduate Studies at King Khalid University, Large Research Project: RGP2/247/46.

Princess Nourah bint Abdulrahman University Researchers Supporting Project: PNURSP2025R719.

Princess Nourah bint Abdulrahman University, Riyadh, Saudi Arabia.

Deanship of Scientific Research at Northern Border University, Arar: NBU-FFR-2025-2899-05.

Deanship of Graduate Studies and Scientific Research at University of Bisha, Fast-Track Research Support Program and Prince Sattam bin Abdulaziz University: PSAU/2024/R/1446.

## Competing Interests

The authors declare that they have no competing interests.

## Author Contributions

- Fatimah Alhayan conceived and designed the experiments, prepared figures and/or tables, and approved the final draft.
- Gaganjot Kaur conceived and designed the experiments, performed the computation work, prepared figures and/or tables, and approved the final draft.
- Sultan Alanazi conceived and designed the experiments, performed the computation work, prepared figures and/or tables, and approved the final draft.
- Mohammed Burhanur Rehman conceived and designed the experiments, analyzed the data, prepared figures and/or tables, and approved the final draft.
- Wahida Mansouri conceived and designed the experiments, analyzed the data, performed the computation work, prepared figures and/or tables, authored or reviewed drafts of the article, and approved the final draft.
- Da'ad Albalawneh performed the experiments, analyzed the data, performed the computation work, authored or reviewed drafts of the article, and approved the final draft.
- Ali Alqazzaz performed the experiments, authored or reviewed drafts of the article, and approved the final draft.
- Hanadi Alkhudhayr performed the experiments, authored or reviewed drafts of the article, and approved the final draft.

## Data Availability

The data is available at GitHub and Zenodo:

- https://github.com/Research-Society/DA-FIS.
- Research Society. (2025). DA-FIS. Zenodo. https://doi.org/10.5281/zenodo.15541493.

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
