# Peer review of "DA-FIS: A high-speed dynamic adaptive fault injection server framework for reliable FPGA-based embedded systems"

_PeerJ Computer Science, doi:10.7717/peerj-cs.2996_

## Round 0.1 · original submission · Major Revisions

Dear authors,

You are advised to critically respond to all comments point by point when preparing an updated version of the manuscript and while preparing for the rebuttal letter. Please address all comments/suggestions provided by reviewers, considering that these should be added to the new version of the manuscript.

Kind regards,
PCoelho

**Language Note:** The review process has identified that the English language must be improved. PeerJ can provide language editing services - please contact us at [email protected] for pricing (be sure to provide your manuscript number and title). Alternatively, you should make your own arrangements to improve the language quality and provide details in your response letter. – PeerJ Staff

Reviewer 1 ·

Basic reporting

The manuscript introduces DA-FIS, a technically robust and contextually significant fault injection framework specifically designed for FPGA-based embedded systems. The proposed solution addresses critical limitations observed in existing methodologies—namely scalability constraints, suboptimal injection speeds, and insufficient adaptability. Experimental results demonstrate that DA-FIS significantly outperforms established frameworks such as BUFIT, SCFIT, and DPR. Nevertheless, although the system architecture and implementation are comprehensively described and empirically validated, the manuscript would benefit from substantial revisions to improve its clarity, analytical depth, and scientific rigor. In particular, the literature review lacks a thorough comparative assessment. The authors should more clearly delineate the technical novelty of DA-FIS by highlighting distinctive architectural innovations, dynamic adaptability mechanisms, and specific engineering enhancements over prior approaches. Moreover, the discussion of related work should not be limited to quantitative performance metrics but should extend to a qualitative analysis of system design trade-offs, integration complexity, and functional scope.

In addition, the manuscript's linguistic quality and structural coherence require careful refinement. Numerous grammatical inaccuracies, stylistic inconsistencies, and imprecise references to figures and tables (e.g., “figure 5,” “this figure 7”) detract from the paper’s overall readability and professional presentation. A comprehensive editorial review is recommended to align with academic publication standards. While the Results and Discussion section presents a range of empirical data, it suffers from repetition and lacks critical interpretation regarding the underlying causes of DA-FIS’s performance gains. The authors should also address the framework’s practical limitations, such as potential architectural bottlenecks, assumptions regarding FPGA configurations, and trade-offs between fault coverage and resource overhead. Furthermore, the manuscript would benefit from a more rigorous statistical analysis of experimental outcomes—such as the inclusion of variance, confidence intervals, or repeated trials—to substantiate performance claims and reinforce result reliability.

Experimental design

The experimental design is appropriate and aligns well with the study's objectives; however, it would benefit from additional detail regarding test repeatability, control conditions, and the rationale behind benchmark selection to enhance the reproducibility and robustness of the evaluation.

Validity of the findings

The findings are supported by comprehensive performance comparisons and demonstrate clear improvements over existing methods; however, the validity could be strengthened by including statistical measures (e.g., standard deviation or confidence intervals) and discussing potential limitations or variability across different FPGA platforms.

Reviewer 2 ·

Basic reporting

This manuscript describes an adaptive fault injection method to evaluate the impact of faults on the FPGA systems. The literature review part was decent. However, the details of the proposed method are completely lacking. There is no explanation about how the LFSR random sequence is used to inject the fault. How the sites for Single Bit Fault and Multi-Bit Fault are selected? What is the relationship between the LFSR-generated random number and the location of the fault? How many bits are in the LFSR? How many bits are there for the FPGA configuration for the sample targets? How does it scale with a real-world application on the FPGA? Will it require an extremely large LFSR?

Experimental design

The figures are mostly flow charts. No design details of the system are illustrated. It is not clear how adaptive the method is and if it is with a different seed of the LFSR. How do different seeds affect the system testing in any meaningful way (as the seed only starts the sequence with a different number and does not increase the number of random numbers that can be generated)?

Validity of the findings

Based on the current writing, it is extremely hard for anyone to reproduce the results and validate the findings independently. The sections on the proposed methodology and Results are extremely weak.

Additional comments

The manuscript does not include sufficient details to make the paper valuable to the readers.

Reviewer 3 ·

Basic reporting

The paper discusses a high-speed approach for fault injection and testing of FPGA designs and shows that their performance and robustness are better than existing solutions.
The key problem with the manuscript is the lack of clarity and details on their framework -- namely, how is the error injected, and how are the specific targets chosen/isolated? Is it to the SRAM layer? Is it using the PCAP/ICAP interface? How do they generate the data formats for writing using PCAP/ICAP? Without these details, it is difficult to appreciate the work done and to faithfully compare against existing solutions.

Experimental design

The LFSR-based model to generate (pseudo) random locations is a nice idea; however, with many solutions, not all resources on the FPGA might be utilised for implementing the final design. How does the LFSR-based approach work in this scenario? Is it pseudo-random within the constraints of the design (based on the floor plan, for instance), or does it ignore the design floor plan completely (in which case, many of the injected errors might not be relevant to the design)?

How do you arrive at the depth of the LFSR?

Why do you need the error injection to have real-time performance? Error-injection testing is done prior to the deployment phase, and there is no practical use for having an error-injection framework once the design is deployed (e.g., satellite in space). The testing phase pre-deployment is elaborate, and there is no clear motivation to have real-time error injection. Error detection, yes - absolutely. But error injection in real-time is not critical.

How are the injection rates measured and compared? Is it measured end-to-end to then compute the average rates, or is it the time to issue the inject command/sequence (with no feedback like posted write commands)?

Validity of the findings

There are details lacking in the paper which makes it hard to validate and appreciate the results.

---

## Round 0.2 · accepted · Accept

Dear authors, we are pleased to verify that you meet the reviewer's valuable feedback to improve your research.

Thank you for considering PeerJ Computer Science and submitting your work.

Kind regards
PCoelho

Reviewer 1 ·

Basic reporting

Thanks for addressing my previous concerns which hopefully enhanced the quality of paper.

Experimental design

Good

Validity of the findings

Good

Additional comments

Good